

# Rural children remain more at risk of acute malnutrition following exit from community based management of acute malnutrition program in South Gondar Zone, Amhara Region, Ethiopia: a comparative cross-sectional study

Dereje B. Abitew[1], Alemayehu Worku[1], Afework Mulugeta[2] and Alessandra N. Bazzano[3]

[1] School of Public Health, Addis Ababa University, Addis Ababa, Ethiopia
[2] School of Public Health, Mekelle University, Mekele, Ethiopia
[3] Department of Global Community Health and Behavioral Sciences, School of Public Health and Tropical Medicine, Tulane University, New Orleans, LA, USA

## ABSTRACT

**Background:** Community-based management of acute malnutrition has been reported effective in terms of recovery rate, but recovered children may be at increased risk of developing acute malnutrition after returning to the same household (HH) environment.

**Objective:** Compare the magnitude and factors associated with acute malnutrition among recovered and never treated children in South Gondar Zone, Amhara Region, Ethiopia.

**Method:** A comparative cross-sectional study was conducted in three districts of South Gondar Zone by tracing 720 recovered and an equal number of age matched children who were never treated for acute malnutrition. Parents were asked to bring children to health post for survey data collection, anthropometric measurements, and edema assessment. Data were collected using a survey questionnaire, entered in to EpiData and analyzed using SPSS v20. Anthropometric indices were generated according to the WHO's 2006 Child Growth Standards using WHO Anthro software version 3.2.2. Bivariate and multivariable logistic regression was utilized. Values with $P < 0.05$ were considered statistically significant and Odds Ratio with 95% CI was used to measure strength of association.

**Result:** A total of 1,440 parents were invited, of which 1,414 participated (98.2% response rate). Mean age in months of children (±SD) was 23.7 (±10.4) for recovered and 23.3 (±10.8) for comparison group. About 49% of recovered and 46% of comparison children were females. A significant difference was observed on magnitude of acute malnutrition between recovered (34.2% (95% CI [30.9–38.0]) and comparison groups (26.7% (95% CI [23.5–30.2]), $P = 0.002$. Factors associated with acute malnutrition among recovered were district of Ebnat (AOR = 3.7; 95% CI [1.9–7.2]), Tach-Gayint (AOR = 2.4; 95% CI [1.2–4.7]); male child (AOR = 1.4; 95% CI [1.0–2.0]); prelactal feeding (AOR = 2.6; 95% CI [1.3 –5.1]); not feeding colostrum (AOR = 1.5; 95% CI [1.1–2.3]); not consuming additional food during

Corresponding author
Dereje B. Abitew,
derejefrae2014@gmail.com

pregnancy/lactation (AOR = 1.6; 95% CI [1.1–2.3]); not given Vitamin A supplement (AOR = 2.1; 95% CI [1.4–3.2]); and safe child feces disposal practice (AOR = 1.7; 95% CI [1.2–2.5]) while district of Tach-Gayint (AOR = 2.5; 95% CI [1.3–4.8]); male child (AOR = 1.5; 95% CI [1.1–2.1]), not feeding colostrum (AOR = 1.7; 95% CI [1.2–2.5]), poor hand washing practice (AOR = 1.6; 95% CI [1.1–2.2]); food insecure HH (AOR = 1.6; 95% CI [1.1–2.4]), birth interval <24 months (AOR = 1.9; 95% CI [1.2–3.2]), and poor access to health facility (AOR = 1.7; 95% CI [1.2–2.4]) were factors associated with acute malnutrition among comparison group.

**Conclusion:** Recovered children were more at risk of acute malnutrition than the comparison group. Nutrition programs should invest in improving nutrition counseling and education; as well as the hygienic practices to protect children against post-discharge relapse of acute malnutrition.

# INTRODUCTION

Under-nutrition is the end result of nutrient deprivation either due to lack of adequate intake or repeated infection (*World Health Organization, 1999*) and is categorized as either chronic or acute malnutrition based on duration of deprivation (*WHO & UNICEF, 2009*). Acute malnutrition is further classified as either moderate acute malnutrition (MAM) or severe acute malnutrition (SAM) based on the degree of malnutrition and the presence of edema (*WHO & UNICEF, 2009*). According to a 2018 report, globally over 49 million children under 5 were wasted and nearly 17 million were severely wasted (*UNICEF, WHO & World-Bank, 2018*) while the magnitude nationally in Ethiopia and in Amhara region was found 7.25% and 7.6% (*Ethiopian Public Health Institute (EPHI), 2019*) and also it was 11.8% in the study area (*Motbainor & Taye, 2019*). Health consequences for children suffering from wasting include weakened immunity and increased risk of death, particularly when wasting is severe (*UNICEF, WHO & World-Bank, 2018*). The mortality rate due to SAM is between 5 and 20 times higher compared with mortality of well-nourished children (*WHO & UNICEF, 2009*; *WHO et al., 2007*). Globally about 1 million children die every year from SAM (*UNICEF et al., 2010*; *World Health Organization, 2013a*), and in Ethiopia about 57% of all under-five deaths are related to malnutrition, of which three-quarters result from complications associated with mild to moderate malnutrition (*Todd et al., 2005*). In addition, about 70% of all childhood mortality in developing countries is due to five major conditions, for which malnutrition (even in mild form) increases the likelihood of mortality up to 56% (*UNICEF et al., 2010*). In order to address nutrition-related public health problems, ten high-impact, nutrition-specific interventions have been identified that, if taken as a package up to 90% coverage, could reduce wasting by 60%, including the management of SAM and MAM as two of the 10 interventions (*Bhutta et al., 2013*).

Children with uncomplicated SAM (WHZ below −3 SD cutoff and/or with mid upper arm circumference (MUAC) cutoff of 115 mm and/or with bilateral edema) and MAM

children (WHZ between −2 and −3 or MUAC between 115 and 125 mm) may be treated in the community setting without requiring admission to a health facility. This process is referred to as Community based Management of Acute Malnutrition (CMAM) (*Collins & Yates, 2003*; *Sphere Project, 2018*) and consists of providing a child with special therapeutic foods, most commonly Ready-to-Use-Therapeutic Food (RUTF) or F75 and F100 milk-based diets, based on severity and a short course of basic oral medication to treat infections (*WHO & UNICEF, 2009*). According to the WHO recommendation, children with SAM are discharged as recovered from treatment when their weight-for-height/length is ≥−2 Z-score and they have had no edema for at least 2 weeks, or mid-upper-arm circumference is ≥125 mm and they have had no edema for at least 2 weeks. Children admitted with only bilateral pitting edema should be discharged from treatment based on whichever anthropometric indicator, mid-upper arm circumference or weight-for-height is routinely used in the local program (*World Health Organization, 2013b*). In the current Ethiopian CMAM program, the admission criteria for children age 6 months to 18 years are: W/H or W/L <70%, or MUAC <110 mm, or presence of bilateral pitting edema. The local national discharge criteria are: W/H ≥85% or WFH ≥−2 Z score for two consecutive weeks (14 days) and no odema for 14 days (if admitted with edema) and the second option is target weight gain for two consecutive visits and no edema for 14 days (if admitted with edema) (*Ethiopian Federal Ministry of Health, 2007*). CMAM programing was established to treat severely acutely malnourished (SAM) children in their communities without being admitted to a health facility (*Collins & Yates, 2003*; *Bhutta et al., 2008*) has been reported effective in terms of the three key performance indicators: recovery, default and death rate (*Sphere Project, 2018*).

In some research findings, the recovery rate has been reported above the Sphere Handbook (*Sphere Project, 2018*) minimum standards of >75% (*Collins, 2007*; *Tekeste et al., 2012*) but post-discharge relapse rates of acute malnutrition (severe and moderate) have been reported as high as 78% (69% for MAM and 9% for SAM) in Bangladesh (*Banerjee, Hoq & Matin, 2016*) 27% (10% SAM and 17% MAM) in Malawi (*Chang et al., 2013*) and 72.1% (34.6% SAM and 37.5% MAM) in Southern Ethiopia (*Tadesse et al., 2018*).

Children discharged after recovery needs to be followed longitudinally to assess improvement over time. The Ethiopian Federal Ministry of Health has recommended one to two months follow up (*Ethiopian Federal Ministry of Health, 2007*), but poor post-discharge follow up has been reported (*Nyirenda & Belachew, 2010*) and based on current findings from the literature (*Stobaugh et al., 2019*), it is plausible that acutely malnourished children who were discharged as recovered may be predisposed to develop acute malnutrition (either MAM or SAM) again after returning to the same household (HH) environment.

Despite emerging data, a gap continues to exist in the evidence base on the differences in acute malnutrition between children who have been treated using CMAM and those who have not. The objective of the current study was to compare the magnitude and factors associated with acute malnutrition between children following exit from CMAM compared with those who were never treated for acute malnutrition in South Gondar Zone of Amhara Region, Ethiopia.

## METHOD AND MATERIALS

### Study area and setting

The study was conducted in South Gondar Zone of Amhara region, Ethiopia. South Gondar Zone is one of 11 administrative zones of Amhara region, Ethiopia. Debretabor is the capital city of the zone situated about 100 km East of Bahir Dar (the capital city of Amhara Region) and 667 km North of Addis Ababa (the capital city of Ethiopia). The Zone has 17 districts, five of which are town administrations. According to the Central Statistical agency of Ethiopia, the 2017/18 population of the Zone was 2,484,929 of which 183, 525 were children 0–4 years old (*CSA, 2013*). In South Gondar, there are one Zonal and three district hospitals, 90 Health Centers, 378 Health Posts, and more than 10 private clinics. According to the Zonal health department annual report, 68% of children under two years of age had received growth monitoring program, and a total of 6,468 SAM children were managed in the health facilities with an overall 96.8% recovery rate (*South Gondar Zone Health Department, 2018*).

### Study design

A comparative cross-sectional study was conducted from 10 November 2017 to 30 January 2018 in South Gondar Zone of Amhara Region, Ethiopia.

### Study population

The study population for the recovered group was children 6–59 months old following exit from CMAM in the randomly selected districts of South Gondar Zone and for the comparison group, the study population were 6–59 months old children who were never treated for SAM in the randomly selected districts of South Gondar Zone.

### Exclusion criteria

Children were excluded if their name was different from the name present in the CMAM registration logbook and/or suffering from any systemic disease.

### Sample size determination and procedure

Since the main objective of this study was to assess any difference in the magnitude of acute malnutrition (wasting) among children following exit from CMAM as compared with children who had never been treated for SAM in the CMAM program, two proportion formula was used. The sample size was calculated using Epi Info version 7.1.3.3 taking percent of acute malnutrition among children following exit from CMAM 78% as described in the literature (*Banerjee, Hoq & Matin, 2016*) and assumption of a 10% difference in percentage of acute malnutrition in the general population (children age 6–59 months) who were never treated for SAM with 95% CL, 80% power, and a ratio of 1:1, the sample size calculated was 656 and when multiplied by a design effect of 2 and including 10% non-response rate, the sample size was 1,440 (720 recovered and 720 from the comparison group).

Regarding the sampling procedure, a multistage sampling technique was used. Among the 17 districts of the Zone, five were town administrations and the remaining 12 were

rural administrative districts. From these rural districts, three districts were selected randomly using lottery method and from these three districts, 10 health centers were selected randomly. In the current Ethiopian health system, a health Centre has five cluster health posts in its catchment area and we considered three to five health posts per health Centre based on the caseload report from the CMAM program. Therefore, 26–43 discharged recovered children were expected per health post. Since this was part of a larger study, a sampling framework developed using the one-year therapeutic multi-chart logbook (from November 2016 to October 2017) to trace 1,290 discharged recovered children was used to select these 720 discharged recovered children taking every second child as a study participant.

An age-matched (±6 months) child who was never treated for SAM but was visiting the health post either for immunization, growth monitoring program, and/or clinical services was selected for the comparison group. Finally, parents of recovered children were contacted by Health Extension Workers and asked to bring children to the health post for survey data collection, anthropometric measurements, and edema assessment.

## Study variables
Dependent variable: Acute malnutrition among children age 6–59 months in South Gondar Zone, Amhara region, Ethiopia.

Independent variables: socioeconomic and demographic, HH hygiene/sanitation, child caring/feeding knowledge/practice, health facility access, and HH food security related variables.

## Data collection tools and measurements
The data collection tools consisted of a checklist and a questionnaire. The checklist was prepared using the Stabilization Centre/outpatient therapeutic program (SC/OTP) multi-chart and registration logbook utilized nationally (*WHO & UNICEF, 2009*; *Ethiopian Federal Ministry of Health, 2007*) to identify and trace discharged recovered children. In addition, an interviewer-administered questionnaire was prepared following a review of pertinent literature (*Ethiopian Public Health Institute (EPHI), 2019*; *Chang et al., 2013*) to address the socioeconomic, demographic, child feeding/caring practices, and health and nutrition-related characteristics of respondents. The questionnaire was prepared in English and then translated to the local language (Amharic) and back to English to check the consistency. Mothers or primary caretakers were interviewed using the questionnaire which took approximately 20–30 min. The data collection took place from 10 November 2017 to 30 January 2018.

To identify history of morbidity of children, mothers were asked about any occurrence of illness during the past 2 weeks and were probed by the enumerator to confirm the nature of the illness based on operational case definitions. Children's vaccination status was checked by observing an immunization card and if not available, mothers were asked to recall child vaccination in as much detail as possible: BCG vaccination was checked if any scar existed on a child's arm.

A total of 15 data collectors who had SAM management training and relevant previous experience in data collection were recruited and trained for 2 days, with training content mainly focusing on anthropometric measurement techniques and on how to administer the questionnaire (*WHO & UNICEF, 2009*; *Ethiopian Federal Ministry of Health, 2007*). The data collectors were closely supervised by the three trained supervisors (public health, nurse or environmental health professionals) and also by the principal investigator.

## Anthropometric measurements

Child MUAC was measured halfway between the olecranon and acromion process using non stretchable tape following the standard procedure and recorded to the nearest 0.1 cm (*WHO & UNICEF, 2009*). Measurements were taken twice and a 0.1 cm variation between the two was accepted as normal. However, repeated measurements were carried upon significantly larger variations according to the stated protocol (*Cogill, 2003*).

In addition, presence of edema was assessed by grasping both feet in the hands with the thumbs on top of the feet and then pressing the thumbs gently for three seconds or a count of 101,102,103 and then releasing the thumbs. It was registered as "0" if no pitting was detected on the feet, recorded as "+" if an indent was detected on feet, "++" if on legs and feet, and "+++" if it included the hands and face according to accepted standards (*WHO & UNICEF, 2009*; *Ethiopian Federal Ministry of Health, 2007*).

## Operational/definition of terms

For the current study, acute malnutrition was defined based on child MUAC <12.5 cm and/or presence of edema, not acutely malnourished otherwise (*WHO & UNICEF, 2009*).

### Good hand washing practice

A respondent was categorized as having good hand washing practice if they reported washing hands at three of four critical times/points (before eating, before preparing food, after defecation, and after cleansing child feces).

### Safe child feces disposal

Respondent's child feces disposal practice was considered safe if the child uses a latrine or if the caregiver placed child feces in a latrine.

### Improved water source

A HH was considered to have improved drinking water if the source was either from a pipe, protected spring, protected well and/or boiled water.

### Currently on FP

A respondent was considered as currently using family planning if she reported having used any family planning methods to avoid pregnancy or extend the interval between births.

### HH food insecurity access scale

Household food insecurity status was determined using the nine item HH Food Insecurity Access Scale question and prior to assigning the food insecurity (access) category, each

frequency of occurrence question was coded as 0 for all cases where the answer to the corresponding occurrence question was "no" and then the four food security categories were created sequentially as recommended by FANTA (*Coates, Swindale & Bilinsky, 2007*). Finally, the HFIA scale category one (I) was considered as food secure and the remaining as food insecure.

## Data management and analysis

The questionnaire was checked manually for completeness and was entered into EpiData version 3.3.2, and exported to SPSS version 20 for analysis. Anthropometric indices were generated according to the WHO's 2006 Child Growth Standards (*World Health Organization, 2007*) using the WHO Anthro software version 3.2.2.

Since the study was comparative, a chi-square test was performed to detect any significant difference in the prevalence of acute malnutrition between the two groups. Univariate analysis was done to see if any association between the outcome and the independent variables and those variables with $P$ value < 0.2 were entered into the final regression model. Variables with $P$ value < 0.05 were considered as predictors of acute malnutrition and strength of association was determined using odds ratio with 95% CIs. The final model was tested for goodness of fit using Hosmer–Lemeshow test and the standard error (SE) values of variables in the final model were checked for multicollinearity.

## Data quality assurance

To assure data quality, data collectors and supervisors were trained for 2 consecutive days on anthropometric measurements techniques based on the stated standard (*WHO & UNICEF, 2009*; *Ethiopian Federal Ministry of Health, 2007*) and were closely supervised by supervisors and investigators. Pre-testing took place in a health post from a non-study district of similar setting to the study sites. Anthropometric tools were checked daily by setting the scale to zero and by measuring a known object (usually IV fluid) to check its accuracy.

## Ethical statement

The protocol was approved by the institutional review board (IRB) of the College of Health Sciences of Addis Ababa University with an approval number of 068/16/SPH, and permission letters were obtained from Regional, Zonal and District health offices of Amhara region, Ethiopia. Informed verbal consent was obtained from all study respondents. Data collectors link children to OTP if child MUAC was <11.0 cm, and/or the child had edema. Nutrition counseling was given to mothers/caretakers after the interview.

## RESULT

### Socioeconomic and demographic characteristics of respondents

Overall there were 1,440 respondents, of which, 1,414 (707 recovered and 707 from the comparison groups) were interviewed (98.2% response rate). All the respondents (i.e., parents/mothers of children) from the two groups were female by gender, rural by residence, and Amhara by ethnicity. The mean age (±SD) in months of children was 23.7

(±10.4) for the recovered and 23.3 (±10.8) for the comparison group. About 44% of recovered and 40% of comparison children age ranged from 12 to 23 months. In addition, 49% of the recovered and 46% of the comparison were female children. The majority (93%) of the respondent's religion in both groups was Orthodox Christians, 94% in the recovered and 90% in the comparison group were currently married. Nearly 60% of respondents in the two groups were unable to read and write. In addition, 81% in the recovered and 77% in the comparison groups were farmers by occupation. About equal (23% of recovered and 25% of comparison group) number of HHs was food insecure. Moreover, about one in six (15%) and one in four (27%) of HHs were in the lowest wealth quintile as indicated in (Table 1).

## Child feeding/caring and housing conditions

About four fifths of children in the two groups (79% of recovered and 81% of comparison group) were reported breastfed and 73% of the mothers in the recovered and 71% in the comparison group reported providing colostrum to the child. The water source for drinking was improved for 56% of the recovered and 58% of the comparison group HHs. About half (50.9% in the recovered and 48.8% in the comparison group) of HHs reported discarding dry human waste into open fields. The majority of children (96% in the recovered and 85% in the comparison group) were vaccinated for measles, and 76% of children in the recovered and 62% in the comparison group received Vitamin A supplementation in the six months preceding the survey (Table 2).

## Prevalence of acute malnutrition

Regarding child nutritional status, a significant difference was observed in the magnitude of acute malnutrition (MUAC < 12.5 cm) between the recovered (34.4%, 95% CI [30.9–38.0]) and the comparison groups (26.7%, 95% CI [23.5–30.2]) $P = 0.002$. From these, 12.0% of the recovered and 9.5% of the comparison group were SAM (MUAC < 11.5 cm). In addition, 66.9% (95% CI [63.3–70.4]) of children in the recovered and 58.3% (95% CI [54.5–61.9]) of children in the comparison group were stunted (HAZ < −2 SD) see (Table 3; Fig. 1).

## Factors associated with acute malnutrition

Regarding factors associated with acute malnutrition, about 28 variables for the recovered, and 16 for the comparison who had $P$ value < 0.2 in the univariate analysis were entered in to the final regression model. Eight variables from the recovered and seven from the comparison group retained their statistical significance at $P < 0.05$. The odds of relapse in the recovered children were more than three times (AOR = 3.67; 95% CI [1.86–7.24]) higher among children who lived at Ebnat and two times (AOR = 2.36; 95% CI [1.20–4.69]) higher among children who lived at Tach-Gayint district compared to those who lived in Lay-Gayint district. The odds of acute malnutrition were about 1.5 times higher in male than female children (AOR = 1.43; 95% CI [1.01–2.01]) and more than two times higher among children who were fed prelacteal food than children who did not receive prelacteal feeding (AOR = 2.56; 95% CI [1.28–5.12]).

**Table 1 Demographic and maternity related.**

| Variable | | Recovered, n (%) | Comparison, n (%) | $X^2$ | P-value |
|---|---|---|---|---|---|
| Respondent age in years | 15–19 | 19 (2.7) | 27 (3.8) | 2.708 | 0.608 |
| | 20–29 | 323 (45.7) | 339 (47.9) | | |
| | 30–39 | 294 (41.6) | 277 (39.2) | | |
| | 40–49 | 67 (9.5) | 61 (8.6) | | |
| | 50+ | 4 (0.6) | 3 (0.4) | | |
| District | Ebnat | 301 (42.6) | 342 (48.4) | 5.538 | 0.063 |
| | Tack Gayint | 308 (43.6) | 267 (37.8) | | |
| | Lay Gayint | 98 (13.9) | 98 (13.9) | | |
| Child age in month | 6–11 | 50 (7.1) | 70 (9.9) | 6.475 | 0.166 |
| | 12–23 | 314 (44.4) | 279 (39.5) | | |
| | 24–35 | 234 (33.1) | 240 (33.6) | | |
| | 36–47 | 73 (10.3) | 73 (10.3) | | |
| | 48–59 | 36 (5.1) | 45 (6.4) | | |
| Sex of child (female) | | 346 (48.9) | 324 (45.8) | 1.373 | 0.241 |
| HH headship (male) | | 662 (93.6) | 618 (87.4) | 15.90 | <0.001 |
| Respondent education status (no formal) | | 580 (82.0) | 556 (78.6) | 2.579 | 0.108 |
| Respondent religion (orthodox) | | 657 (92.9) | 654 (92.5) | 0.094 | 0.759 |
| Marital status (currently married) | | 661 (93.5) | 633 (89.5) | 7.139 | 0.008 |
| HH family size (>=5) | | 400 (56.6) | 373 (52.8) | 2.08 | 0.149 |
| Total under 5 children (2+) | | 166 (23.5) | 153 (21.6) | 0.684 | 0.408 |
| Respondent occupation (farming) | | 573 (81.0) | 542 (76.7) | 4.076 | 0.043 |
| HH currently on food aid (yes) | | 189 (26.7) | 217 (30.7) | 2.709 | 0.100 |
| Own at least one HH effect* (yes) | | 135 (19.1) | 183 (25.9) | 9.347 | 0.002 |
| HH own agricultural land (yes) | | 655 (93.3) | 596 (88.8) | 26.78 | <0.001 |
| HH own farm animals (yes) | | 654 (92.5) | 590 (83.5) | 27.39 | <0.001 |
| HH food insecure (yes) | | 164 (23.2) | 178 (25.2) | 0.756 | 0.358 |
| Decision maker on HH expenditure (together) | | 300 (42.4) | 241 (34.1) | 10.44 | 0.001 |
| Attend ANC during this pregnancy (yes) | | 564 (79.8) | 595 (84.2) | 4.598 | 0.032 |
| Place of delivery (home) | | 244 (34.5) | 236 (33.4) | 0.202 | 0.653 |
| Additional food consumption at pregnancy/lactation (yes) | | 432 (61.1) | 407 (57.6) | 1.832 | 0.176 |
| Currently on family planning (yes) | | 564 (79.8) | 468 (66.2) | 33.06 | <0.001 |
| Birth interval in month (<24 month) | | 104 (14.8) | 82 (11.6) | 2.996 | 0.083 |
| Birth order (3+) | | 412 (58.3) | 387 (54.7) | 1.789 | 0.180 |
| Home to HP walk in minute (<=30) | | 305 (43.1) | 383 (54.2) | 17.22 | <0.001 |
| Wealth quintile (lowest) | | 105 (15.0) | 183 (26.1) | 29.21 | <0.001 |

Notes:
HH, household; ANC, Antenatal care; HP, health post.
* Radio, TV, mobile, refrigerator.

Similarly, the odds of acute malnutrition in children from the comparison group were 2 times (AOR = 2.49; 95% CI [1.29–4.81]) higher among children who lived at Tach-Gayint compared to those who lived at Lay-Gayint district. The odds of acute malnutrition were 1.5 times higher in male than female children (AOR = 1.47; 95% CI [1.02–2.11]) and about

Table 2 Child feeding/caring and housing conditions.

| Variables | Recovered, n (%) | Comparison, n (%) | $X^2$ | P-value |
|---|---|---|---|---|
| BF initiation immediately within an hour after birth | 447 (63.2) | 461 (65.5) | 0.784 | 0.376 |
| Colostrum provided (yes) | 516 (73.0) | 501 (71.2) | 0.580 | 0.446 |
| Prelactal feeding (yes) | 47 (6.6) | 35 (5.0) | 1.864 | 0.172 |
| Child currently on breastfeeding (yes) | 559 (79.1) | 572 (80.9) | 0.747 | 0.388 |
| Complementary feeding as WHO recommended (2–4 times/day) | 614 (86.8) | 635 (89.8) | 3.026 | 0.082 |
| Prepare food to children other than family diet (yes) | 431 (61.0) | 391 (55.3) | 4.649 | 0.031 |
| Trained in child food preparation (yes) | 406 (57.4) | 372 (52.6) | 3.303 | 0.068 |
| Child vaccinated for measles (yes) | 678 (95.9) | 602 (85.1) | 47.62 | 0.0001 |
| Vitamin A received in the last 6 months (yes) | 536 (75.8) | 438 (62.0) | 31.69 | 0.0001 |
| Deworming given in the past 6 months (yes) | 150 (21.2) | 140 (19.8) | 0.434 | 0.510 |
| History of diarrhea last 2 weeks (yes) | 61 (8.6) | 75 (10.6) | 1.595 | 0.207 |
| HH water source type (improved) | 397 (56.2) | 408 (57.7) | 0.349 | 0.555 |
| HH latrine type (traditional pit latrine) | 625 (88.4) | 609 (86.1) | 1.630 | 0.202 |
| HH dry waste disposed (open fields) | 360 (50.9) | 345 (48.8) | 0.636 | 0.425 |
| HH child feces dispose practice (safe) | 388 (54.9) | 424 (60) | 3.494 | 0.053 |
| Hand washing practice (good) | 400 (56.6) | 388 (54.9) | 0.143 | 0.521 |
| Often wash hands with (water only) | 352 (49.8) | 364 (51.5) | 0.777 | 0.678 |
| Time to water source (<30 min walk) | 280 (17.3) | 309 (43.7) | 2.447 | 0.118 |
| Other family member on SAM treatment (yes) | 19 (2.7) | 6 (0.8) | 6.882 | 0.009 |
| Average OTP treatment stay in weeks (8..0 ± 1.7) | | | | |
| Average duration after recovery in month (4.8 ± 1.9) | | NA | | |
| Duration since recovery in month (<=3) | 234 (33.1) | NA | | |

Note:
  SAM, severe acute malnutrition; OTP, outpatient therapeutic program; NA, not applicable.

1.6 times higher among children in food insecure than food secure HHs (AOR = 1.59; 95% CI [1.06–2.38]). In addition, the odds of acute malnutrition were two times higher among children with birth interval of <24 months than those with birth interval above >=24 months (AOR = 1.92; 95% CI [1.16–3.19]), and 1.5 times higher among children whose mothers had practiced hand washing only after one or two of the key points compared to those who wash hands more frequently (AOR = 1.55; 95% CI [1.08–2.21]) (Table 4).

## DISCUSSION

In the current study, the prevalence of acute malnutrition was higher in the recovered than the comparison group.

Some health and nutrition organizations have reported that recovery rates from malnutrition are above the Sphere (Sphere Project, 2018) minimum standard of >75% (Collins, 2007; Tekeste et al., 2012; Efrem et al., 2010) indicating good progress, but recovered children may still be in a state of heightened risk for acute malnutrition. One possible issue is lack of follow up intervention after discharge as recovered. The Ethiopian Federal Ministry of Health has recommended linking to a supplementary

**Table 3 Prevalence of acute malnutrition.**

| Indicators | | Recovered | | Comparison group | | P value |
|---|---|---|---|---|---|---|
| | | n (%) | 95% CI | n (%) | 95% CI | |
| MUAC (cm) | Normal (≥12.5) | 464 (65.6) | 62.0–69.1 | 518 (73.3) | 69.8–76.5 | 0.002 |
| | Wasted (<12.5) | 243 (34.4) | 30.9–38.0 | 189 (26.7) | 23.5–30.2 | |
| WHZ | Normal (≥−2SD) | 484 (68.5) | 64.9–71.9 | 542 (76.7) | 73.4–79.7 | <0.001 |
| | Wasted <−2SD) | 223 (31.5) | 28.1–35.1 | 165 (23.3) | 20.3–26.6 | |
| HAZ | Normal (≥−2SD) | 234 (33.1) | 29.6–36.7 | 295 (41.7) | 38.1–45.5 | 0.001 |
| | Stunted (<−2SD) | 473 (66.9) | 63.3–70.4 | 412 (58.3) | 54.5–61.9 | |
| WAZ | Normal (≥−2SD) | 345 (48.8) | 45.1–52.6 | 430 (60.8) | 57.1–64.4 | <0.001 |
| | Underweight (<−2SD) | 362 (51.2) | 47.4–54.9 | 277 (39.2) | 35.6–42.9 | |

Note:
MUAC, mid upper arm circumference; WHZ, Weight for Height Z score; HAZ, Height for age Z score; WAZ, Weight for Age Z score.

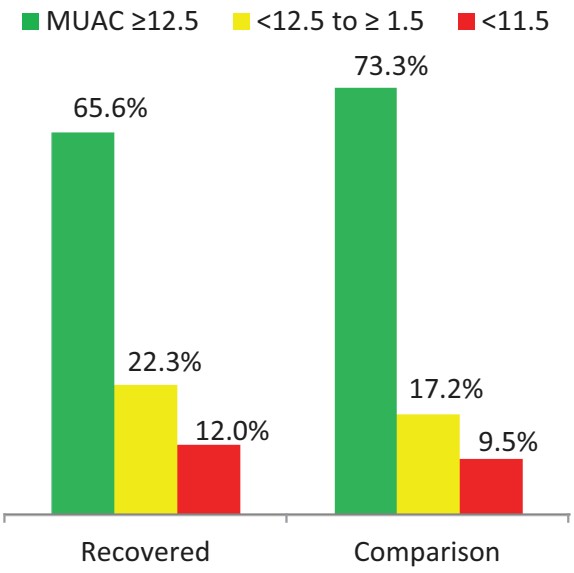

**Figure 1 Acute malnutrition in the two groups.**

feeding program for another 4 months following exit from therapeutic programs (*Ethiopian Federal Ministry of Health, 2007*). But in the current study, none of the children were linked to supplementary feeding programs, so were likely at high risk of relapse. Studies have reported that children who were treated for 12 weeks after fulfilling discharge criteria had lower rates of SAM (*Trehan et al., 2015*), similarly children who continued supplementary feeding longer than the required time remained well-nourished over the subsequent year (*Chang et al., 2013*). An additional potential factor could be due to the changes in the eating habits of children. Children who were in the program receiving therapeutic food (ready to use therapeutic food (RUTF), F100, and/or F75) may not be willing to return to receiving family food as frequently as before admission to the program. In addition, these recovered children may go through repeated episodes of treatment with RUTF (*WHO et al., 2007*) and such increased consumption could lead them to have a

**Table 4 Factors associated with acute malnutrition.**

| Variables | | Acute malnutrition in discharge as recovered children | | | | Acute malnutrition in comparison group | | | |
|---|---|---|---|---|---|---|---|---|---|
| | | Yes | No | COR [95% CI] | AOR [95% CI] | Yes | No | COR [95% CI] | AOR [95% CI] |
| District | Ebnat | 125 | 176 | 4.26 [2.31–7.85] | 3.67 [1.86–7.24] | 80 | 262 | 1.69 [0.92–3.09] | 1.65 [0.86–3.14] |
| | Tach-Gayint | 104 | 204 | 3.06 [1.66–5.65] | 2.36 [1.20–4.69] | 94 | 173 | 3.01 [1.64–5.50] | 2.49 [1.29–4.81] |
| | Lay Gayint | 14 | 84 | 1 | 1 | 15 | 83 | 1 | 1 |
| Sex | Male | 133 | 228 | 1.25 [0.92–1.71] | 1.43 [1.01–2.01] | 113 | 270 | 1.37 [0.97–1.92] | 1.47 [1.02–2.11] |
| | Female | 110 | 236 | 1 | 1 | 76 | 248 | 1 | 1 |
| Prelacteal feeding | No | 215 | 445 | 1 | 1 | | | | |
| | Yes | 28 | 19 | 3.05 [1.67–5.59] | 2.56 [1.28–5.12] | | | | |
| Vitamin A supplement in past 6 months | No/don't | 91 | 80 | 2.87 [2.02–4.10] | 2.13 [1.42–3.19] | | | | |
| | Yes | 152 | 384 | 1 | 1 | | | | |
| Child feces disposal | Unsafe | 99 | 220 | 1 | 1 | | | | |
| | Safe | 144 | 244 | 1.31 [0.96–1.80] | 1.73 [1.21–2.48] | | | | |
| Additional food at pregnancy/lactation | No | 124 | 151 | 2.16 [1.57–2.97] | 1.58 [1.10–2.28] | | | | |
| | Yes | 119 | 313 | 1 | 1 | | | | |
| Colostrum feeding | No | 82 | 109 | 1.66 (1.18–2.34) | 1.51 (1.01–2.26) | 74 | 129 | 1.97 [1.38–2.81] | 1.66 [1.23–2.45] |
| | Yes | 161 | 355 | 1 | 1 | 113 | 388 | 1 | 1 |
| Respondent age in years | <=30 | 97 | 245 | 1 | 1 | | | | |
| | >30 | 146 | 219 | 1.68 [1.23–2.21] | 1.63 [1.15–2.31] | | | | |
| HH food security | No | | | | | 68 | 110 | 2.08 [1.45–3.00] | 1.59 [1.06–2.38] |
| | Yes | | | | | 121 | 408 | 1 | 1 |
| Good hand washing practice | No | | | | | 100 | 219 | 1.53 [1.10–2.14] | 1.55 [1.08–2.21] |
| | Yes | | | | | 89 | 299 | 1 | 1 |
| Birth interval in month | <24 | | | | | 31 | 51 | 1.80 [1.11–2.91] | 1.92 [1.15–3.19] |
| | >=24 | | | | | 158 | 467 | 1 | 1 |
| Home to HP by foot walk in minute | <=30 | | | | | 83 | 300 | 1 | 1 |
| | >30 | | | | | 106 | 218 | 1.76 [1.26–2.46] | 1.66 [1.16–2.37] |

modification of taste preference towards sweet or fatty foods (*Rozin & Vollmecke, 1986*). In addition, even though the long-term impact of consumption of RUTF on the gut microbiome of SAM children is not yet fully understood, there is some evidence that microbiota are easily manipulatable by dietary changes (*Alcock, Maley & Aktipis, 2014*) and this has been implicated as a causal factor of kwashiorkor (*Smith et al., 2013*).

Regarding the factors identified to be associated with acute malnutrition, eight variables (district of residence, sex of the child, prelacteal feeding, colostrum feeding, Vitamin A supplementation, child feces disposal practice, respondent age in years, and consumption of additional food during pregnancy/lactation) in the recovered group and seven variables (district of residence, sex of the child, colostrum feeding, HH food security status, home to health post distance, respondent hand washing practice, and birth interval) in the comparison groups were identified as predictors of acute malnutrition. Three of these predictors (district of residence, sex of the child, and colostrum feeding) were common in both groups.

In the current study, sex of the child was the factor associated with acute malnutrition. Male children both in the recovered and in the comparator group were more acutely malnourished than female children. This finding was in line with studies conducted in Malawi (*Chang et al., 2013*; *Trehan et al., 2015*), Sierra Leone (*Amanda et al., 2015*), Tanzania (*Mgongo et al., 2017*), Nigeria (*Udoh & Amodu, 2016*), Burkina Faso (*Poda, Hsu & Chao, 2017*) and Ethiopia (*Demissie & Worku, 2013*; *Taye, Wolde & Seid, 2016*). One possible factor could be the typical activities of boys in to be away from home performing tasks leading to missed meals. A recent study has also noted that boys health may be more influenced by environmental stressor and diarrhea (*Kumi-Kyereme & Amo-Adjei, 2016*).

District of residence was a factor associated with acute malnutrition both in the recovered and comparison groups. Other studies reporting that geographic variation was associated with acute malnutrition in Bangladesh in relation to Eastern vs. Southern district (*Chowdhury et al., 2016*) and in Sylhet vs. Dhaka division (*Talukder, 2013*) as well as in Zambia between Northern vs. Western provinces (*Nzala et al., 2011*). This could be linked to malaria transmission by geographic area (*Gone et al., 2017*), or The difference could also be related to child food preparation practice (as one third of mothers from Lay-Gaying vs. one 10th from Ebnat). A study in Afar region, Ethiopia found that children's food wasn't prepared separately from family food were more acutely malnourished (*Seid, Seyoum & Mesfin, 2017*).

Children both in the recovered and in the comparison groups who were not given colostrum were found more acutely malnourished than those who were given colostrum. This is because colostrum has all the essential nutrients and immunoglobulin that are important in disease prevention. The finding was in line with studies from India (*Mishra et al., 2014*) and Ethiopia (*Bantamen, Belaynew & Dube, 2014*). Practice of colostrum feeding may be related to maternal education/counseling on child nutrition/health as those who had good nutrition knowledge had better infant feeding practices (*Guldan et al., 2000*; *Daba & Ersado, 2015*) and to have more well-nourished children (*Appoh & Krekling, 2005*). The practice of EBF may also linked, as exclusively breast fed children were less acutely malnourished than their counterparts (*Egata, Berhane & Worku, 2014*; *Awoke, Ayana & Gualu, 2016*).

Children who were given prelacteal feeding were more acutely malnourished than those who were not. The finding echoes other reports from India (*Mishra et al., 2014*; *Ambadekar & Zodpey, 2017*), and Ethiopia (*Demissie & Worku, 2013*). Prelacteal feedings disrupts the feeding of colostrum, the practice of EBF, and increase likelihood of other foods being introduced before 6 months, as has been reported in Ethiopia and Vietnam (*Tewabe et al., 2016*; *Qadri & Srivastav, 2017*). This also results in enteric infections and environmental enteropathy due to consumption of unsafe water or liquids (*Sanghvi, Mehta & Kumar, 2014*).

The odds of relapse were higher among children not given high dose Vitamin A supplementation in the 6 months preceding the survey. This may indicate low receipt of, or access to nutrition services such as distribution of Vitamin A capsule and has also been noted in Hawassa, and Afar region, Ethiopia (*Seid, Seyoum & Mesfin, 2017*; *Bisrat &*
*Kulkarni, 2017*). The protective role of Vitamin A in promoting and regulating activities in both the innate and adaptive immune system (*Huang et al., 2018*) enhancing immune function is the rationale for recommended supplementation every 6 months (*World Health Organization, 2011*).

International recommendations for pregnant and lactating women include at least one extra serving of food per day to meet extra caloric needs (*WHO & UNICEF, 2003*). The finding was in agreement with a study conducted in East Gojjam, Ethiopia describing more wasting among children whose mothers' were not able to consume extra food during pregnancy (*Awoke, Ayana & Gualu, 2016*). Undernourished (BMI < 18.5) mothers are more likely to have undernourished children (*Chowdhury et al., 2016*; *Ambadekar & Zodpey, 2017*).

Safe child feces disposal is important in reducing diarrheal diseases and malnutrition, but in the current study, more acutely malnourished children were identified in HHs who reported practicing safe child feces disposal than in those who dispose in to open field/bush. A study in Zambia noted that children who always used latrine were more wasted than those whose feces was buried (*Nzala et al., 2011*). Yet a study in India reported that those children who utilize latrine were less acutely malnourished (*Ambadekar & Zodpey, 2017*). Authors in Indonesia reported that those who disposed of child stool safely had fewer episodes of diarrhea (*Cronin et al., 2016*). Availability of latrine facilities and/or assuming safe child feces disposal is therefore not a guarantee unless it is been practiced in an optimal condition.

Unsurprisingly, the odds of acute malnutrition among children in food insecure HHs were higher than in food secure HHs. The finding was in agreement with studies conducted in India (*Burza et al., 2016*), Nigeria (*Ajao et al., 2010*) and Ethiopia (*Bisrat & Kulkarni, 2017*) Acute malnutrition in food insecure HHs can also be linked to inadequate intake of diversified foods as studies have reported that consumption of low dietary diversity food (≤3 food groups) as being associated with acute malnutrition (*Miskir et al., 2017*; *Dodos et al., 2018*; *Frozanfar et al., 2016*). Other studies also pointed to the role of low socioeconomic status or monthly income in food insecurity (*Appoh & Krekling, 2005*; *Frozanfar et al., 2016*) which directly or indirectly reduces the HH purchasing power, and thus, reduces access to food.

Odds of acute malnutrition were higher among children whose mothers had practiced hand washing only after one or two of the key points compared to those who washed hands more frequently. The World Health Organization (WHO) recommends washing hands at four key time points: before eating, before preparing food, after defecation, and after disposal of child feces (*WHO, UNICEF & USAID, 2015*). Our finding are similar to other studies where caregivers who washed hands regularly were less likely to have acutely malnourished children (*Bantamen, Belaynew & Dube, 2014*; *Ambadekar & Zodpey, 2017*; *Dodos et al., 2018*). Hand washing reduces the risk of contamination by excreta and thereby transmission of pathogens (*Brown, Cairncross & Ensink, 2013*). One of the strength of this study is that this is the first cross-sectional study employing a comparative (control) group in Ethiopia to better understand if relapse rate differed following exit compared with non-treated cohorts. Despite standardization of anthropometric

instruments, intensive training, and close supervisions, misclassification of children's nutritional status due to measurement error is potentially a limitation. Children treated for SAM before could be considered as never treated due to caretakers recall bias and also their children immunization status. Inclusion of children attending for clinical services could overestimate the burden of wasting. In addition, HH food insecurity status may be underestimated, as the data collection time was in the harvest season (October 2017 to January 2018).

## CONCLUSION AND RECOMMENDATION

In the present study, recovered children were more at risk of acute malnutrition than the comparison group. Three variables, (district of residence, sex of the child and not feeding colostrum) were common predictors in the two groups. Understanding the factors associated with acute malnutrition may help improve in prevention and management of future post discharge relapse. Nutrition programs should invest in improving nutrition counseling and education, especially focusing on IYCF practices, as well as the hygienic practices to protect children against post-discharge relapse of acute malnutrition.

## ACKNOWLEDGEMENTS

Our appreciation goes to the Amhara Regional Health Bureau, South Gondar Zone Health Department, and all three district health office heads & nutrition focal persons. We especially thank the study participants, supervisors and data collectors.

### Funding

This study was funded by Addis Ababa University, School of Public Health who provided money for data collection. The funders had no role in study design, data collection and analysis, decision to publish, or preparation of the manuscript.

### Grant Disclosures

The following grant information was disclosed by the authors:
Addis Ababa University, School of Public Health.

### Competing Interests

The authors declare that they have no competing interests.

### Author Contributions

- Dereje B. Abitew conceived and designed the experiments, performed the experiments, analyzed the data, prepared figures and/or tables, authored or reviewed drafts of the paper, and approved the final draft.
- Alemayehu Worku conceived and designed the experiments, performed the experiments, analyzed the data, prepared figures and/or tables, authored or reviewed drafts of the paper, and approved the final draft.

- Afework Mulugeta conceived and designed the experiments, performed the experiments, analyzed the data, prepared figures and/or tables, authored or reviewed drafts of the paper, and approved the final draft.
- Alessandra N. Bazzano performed the experiments, analyzed the data, prepared figures and/or tables, authored or reviewed drafts of the paper, and approved the final draft.

## Human Ethics

The following information was supplied relating to ethical approvals (i.e., approving body and any reference numbers):

The institutional review board (IRB) of the College of Health Sciences of Addis Ababa University approved the study (068/16/SPH).

## Data Availability

The raw data is available as Supplemental Files.

## Supplemental Information

Supplemental information for this article can be found online at http://dx.doi.org/10.7717/peerj.8419#supplemental-information.

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
