# Peer review of "Rural children remain more at risk of acute malnutrition following exit from community based management of acute malnutrition program in South Gondar Zone, Amhara Region, Ethiopia: a comparative cross-sectional study"

_PeerJ, doi:10.7717/peerj.8419_

## Round 0.1 · original submission · Major Revisions

· Academic Editor

Major Revisions

This study attempts to generate an evidence for burden of acute malnutrition among those treated in CMAM and recovered compared to those who were never treated. It is also understood that this work is part of a larger study. Authors have presented a fairly structured manuscript.
Reviewers have pointed out few corrections in content as well as formatting. These comments needs to be addressed.
In addition, please review following comments and respond accordingly.
• Sample size assumptions- Calculations were based on higher proportions reported. However what study have actually found is almost half of the expected. How does this influence internal validity for the study? Present a power calculation for current findings.
• Study population- CMAM program runs for maximum of 12 weeks, and is meant for children between 6 to 59 months. Then inclusion criteria of 9 to 59 months wouldn’t be better?
• Who were the children admitted in CMAM, whether they were SAM (MUAC <11 ) or both SAM and MAM (MUAC <12.5) ? What was the discharge criteria? Was it MUAC >12.5 or MUAC >11 ? or Was it WHZ >-2? or WHZ >-3?
• Operational definition Acute Malnutrition in this study is MUAC <12.5, please examine how many from recovered group were having MUAC >11 and <12.5? If there were any such children then, selection bias is playing role in higher proportion of Acute Malnutrition among those ‘recovered’.
• Recovered group is of such children who were diagnosed as SAM once, treated and recovered. Which means they were definitely not having acute malnutrition at the time of dicharge. For other group, do we know if they were ever been diagnosed as SAM or referred for CMAM program? Population base from which they are coming, is it same as those of treated in CMAM programme?
• When data is examined, it looks that around one third of recovered children were discharged within three months and most were discharged within six months. This follow up interval after discharge would influence prevalence of acute malnutrition. Please examine this variable and its influence on outcome.
• Results should be presented with Conceptual Framework for Malnutrition which classifies causes as Underlying, Intermediate and Immediate. It will help readers comprehend better.
• Data Management and Analysis
• Why predictors of acute malnutrition were determined separately among those recovered and those never treated?
• For logistic regression, How is multicollinearity checked?
• Please describe model diagnostic and fitting information. What were results of Hosmer Lemeshow Test? How much was Pseudo-R square?
• You can use a graph to represent results of logistic regression (OR and 95% CI), wherein Odds Ratio will be on X-axis , and each variable with its OR and 95% CI could be presented one above the other. (A graph analogues to forest plot). This graph creation is optional.
• Also, since malnutrition is once corrected among those recovered, how much weightage should be given to the determinants at the time of birth needs to be thought upon.
• There are many variables which actually relates with Socio-economic position, whether Wealth Index or some other composite measure is constructed? Use of such composite measure may reduce number of variables to be entered in logistic regression and may change results. Please explore this possibility.
• Discuss limitations- There is possibility of differential recall for some variables among those previously treated for SAM owing to counselling and education they have received.

Reviewer 1 ·

Basic reporting

• The write-up is clear and understandable.
• The format is well addressed and coherent.
• Literature well referenced, however old references have been cited (Line 54),
• Acronyms are not fully written for the first time SAM and MAM (Line 56), SPHERE hand book (Line 67),
• Results are presented in subtitle which is good to catch up ideas easily.
• Factors associated with acute malnutrition (Line 278, particularly from 279-283) looks long and vague. Pls try shortening.
• Maintain tables number order based on the position in the main document
• In all tables, good to indicate chi-square and p-values)
• The missing cells in Logistic regression table for certain variables are not clear. Why is that? How do we compare the recovered and comparison groups?
• The figure is very attractive and speaks properly.

Experimental design

• A well-defined research questions and well planned to full fill the gaps of the area.
• The research seems to be original in nature, since the authors explained that the program is intervened in the area recently. So they explored the impact of the program.
• The methods are very detail and tried to clarify all step by step. However, some parts are too detail, like data management and analysis. If too long readers may not be interested to read by looking the large number of pages.
• Operational definition needs to be referenced.
• In the statistical analysis, I recommend the ANOVA for the two groups. Explain why you didn’t consider?
• Rigorous investigation performed to a high technical & ethical standard. Authors indicated the ethical procedures with reference letter. I appreciated refereeing children to OTP if child MUAC was <11.0cm, and/or the child had edema.
• Why respondents from both groups only female by their gender? Was it intentional or any other justification? (Line 251-252)
• A very long discussion which explored everything in detail. It has to shorten as much as possible within 2-4 pages with the most relevant ones.

Validity of the findings

• Recall to immunization status of the child exposed to some level of bias of recall. (Line 142-143).
• Conclusions are well stated, linked to original research question, but in line 492, “take home ration with strict follow up” is that a sustainable and feasible intervention?
• Your conclusion should be focused based on the variables you identified as significant to SAM in recovered children.

Additional comments

I found this study is a well comprehensive and well explored which is relevant for developing countries where nutrition is their big concerns.

Annotated reviews are not available for download in order to protect the identity of reviewers who chose to remain anonymous.

·

Basic reporting

Generally the manuscript English language utilization is good but needs minor correction on the following point:
a. Sentence from line 57-61 is long and difficult to understand
b. Sentence on line 73 needs grammatical correction, better if you rewrite in such away ‘children discharged after recovery needs to be followed longitudinally to assess improvements over time’
c. A sentence from line 130-134 need to be break down into two for more clarity.
On line 262 replace 4/5th by ‘four fifth’
d. A paragraph from line 309-339 is very long with multiple idea which needs reorganization.
e. At the end of line 387 put full stop.
The context of is well described even though some of the reference were very old even beyond 10 year ago. In addition it is good to incorporate the magnitude of acute malnutrition in the general population (first episode which is the comparative population in your study) and nothing was mentioned about the factors. The availability of raw data made the authors genuine for the scientific world and incorporating short description will made it easily understandable.
Tables:
- Avoid back ground coloring
- Correct the table numbering since all were labeled as ‘table 1’
- The variables ANC, palace of birth, FP, Food consumption at pregnancy & lactation are not in line with table title ‘socioeconomic and demographic….’
- Incorporate the population (children age 6-59 mothers) on the title of the table of associated factors.
Figure: put the title of the figure below the figure.

Experimental design

The method section is well constructed but the following improvements are needed:
a. Indicate the study period
b. Provide the source for the information mentioned under the study setting
c. Avoid the title from the study design since it is repetition
d. Why don’t you consider previous study magnitude of acute malnutrition among the general population rather than taking 10% mean difference during sample size determination?
e. Why the urban district was totally removed during the sampling procedure? If they have to be removed the title and study population should be corrected considering only the rural district.
f. The last paragraph under sample size determination and procedure is better to under data collection procedure.

Validity of the findings

Result:
a. Is child age included under multivariable binary logistic regression as stated on line 280 & 281, if so why since p>0.2
Discussion:
a. The first statement is not relevant better to be removed.
b. For the statement from line 309-312 put the reference.
c. On line 315&316 it was stated that as ‘none of the recovered children were linked to supplementary feeding programs’ but nothing was mentioned about it in the result section?
d. Maternal educational attainment, preparing child food separately and receiving child food preparation training were stated as possible explanation district difference of acute malnutrition. Why?
e. Incorporate the possible explanation for colostrum feeding and acute malnutrition
f. How safe child feces disposal increase the odds of acute malnutrition? Put possible explanation.
g. Household food insecurity was mentioned as one of the factors but not well discussed. In addition socioeconomic status was indicated as a possible explanation while noting was mention in the result section. How?
Conclusion & recommendation: under this linking to supplementary feeding program, home take ration with strict follow up were recommended whereas nothing were stated in the result.

---

## Round 0.2 · Minor Revisions

· Academic Editor

Minor Revisions

The manuscript is now improved from its previous version.

Please go through the remaining reviewer suggestions and revise accordingly.

·

Basic reporting

No comment

Experimental design

Thank you for considering most of the comment forwarded previously. But still I do have doubt on the generalizability of the study finding to both the urban and rural area. Because your sample population considered only the rural area due to the mentioned reason on your response. So that it better to indicate as the study only covered rural area in the title and population for the seek of generalizability.

Validity of the findings

The authors tried their best to improve the study in meeting the standard. Here I do have some issues:-
1. The first one is with regard the criteria used to select variables for the final model, in which both statistical and theoretical criteria. So that the authors followed double standard in selecting the variables, for instance age was selected based on the their theoretical knowledge where as the rest were selected using statistical criteria(p-value<0.2). This is introducing error or bias as far as my knowledge. Therefore, they should follow similar criteria in selecting all variables.
2. The second one is with regarding the conclusion and recommendation, in which the authors need to conclude and recommend based on the finding of the study. So that better to remove issus mentioned about supplementary feeding program and home take ration.

---

## Round 0.3 · accepted · Accept

· Academic Editor

Accept

The manuscript is now well structured and queries are well sorted out.